# Unraveling the Ancient Introgression History of *Xanthoceras* (Sapindaceae): Insights from Phylogenomic Analysis

**DOI:** 10.3390/ijms26041581

**Published:** 2025-02-13

**Authors:** Jian He, Mingyang Li, Huanyu Wu, Jin Cheng, Lei Xie

**Affiliations:** State Key Laboratory of Efficient Production of Forest Resources, Beijing Forestry University, Beijing 100083, China; lmylmy2220103@bjfu.edu.cn (M.L.); wuhuanyu22@bjfu.edu.cn (H.W.); chengjin@bjfu.edu.cn (J.C.)

**Keywords:** ancient introgression, cyto-nuclear discordance, sapindaceae, transcriptome phylogenomics, *Xanthoceras*

## Abstract

Ancient introgression is an infrequent evolutionary process often associated with conflicts between nuclear and organellar phylogenies. Determining whether such conflicts arise from introgression, incomplete lineage sorting (ILS), or other processes is essential to understanding plant diversification. Previous studies have reported phylogenetic discordance in the placement of *Xanthoceras*, but its causes remain unclear. Here, we analyzed transcriptome data from 41 Sapindaceae samples to reconstruct phylogenies and investigate this discordance. While nuclear phylogenies consistently placed *Xanthoceras* as sister to subfam. Hippocastanoideae, plastid data positioned it as the earliest-diverging lineage within Sapindaceae. Our coalescent simulations suggest that this cyto-nuclear discordance is unlikely to be explained by ILS alone. HyDe and PhyloNet analyses provided strong evidence that *Xanthoceras* experienced ancient introgression, incorporating approximately 16% of its genetic material from ancestral subfam. Sapindoideae lineages. Morphological traits further support this evolutionary history, reflecting characteristics of both contributing subfamilies. Likely occurring during the Paleogene, this introgression represents a rare instance of cross-subfamily gene flow shaping the evolutionary trajectory of a major plant lineage. Our findings clarify the evolutionary history of *Xanthoceras* and underscore the role of ancient introgression in driving phylogenetic conflicts, offering a rare example of introgression-driven diversification in angiosperms.

## 1. Introduction

Interspecific gene flow is a captivating evolutionary phenomenon that has intrigued biologists for centuries due to its profound impact on the diversification and adaptability of living organisms [1,2,3,4,5,6]. By facilitating gene flow between distinct lineages, admixture can introduce novel allele combinations that drive morphological innovation, ecological shifts, and even the emergence of entirely new taxa [7,8,9,10]. Its significance has been demonstrated across a remarkable array of plant and animal lineages, underscoring interspecific gene flow’s role as a powerful evolutionary force that reshapes phylogenetic relationships and contributes to the richness of life’s diversity [11]. However, most documented interspecific gene flow cases involve closely related species [12,13,14], while introgression between genera or distantly related lineages remains exceedingly rare [15,16]. One key indicator of ancient gene flow events is cyto-nuclear discordance, where incongruences between plastid and nuclear gene trees often point to past introgression or reticulate evolutionary processes [17,18,19,20].

An intriguing example of pronounced cyto-nuclear discordance is observed in *Xanthoceras*, the sole genus of the monotypic subfam. Xanthoceroideae (Sapindaceae), which contains a single species, *Xanthoceras sorbifolium* (commonly known as the goldenhorn or yellowhorn). This deciduous shrub or small tree, characterized by its reddish-brown branches, pinnately compound leaves, and large flowers and fruits, thrives in arid regions of northern China and the Korean Peninsula [21].

The phylogenetic placement of *Xanthoceras* within Sapindaceae has long been debated. The family comprises four subfamilies: Xanthoceroideae, Dodonaeoideae, Hippocastanoideae, and Sapindoideae [22,23,24,25,26,27,28,29,30,31,32]. While chloroplast-gene-based analyses often place subfam. Xanthoceroideae as the earliest-diverging lineage [22,23,24,25,26,27,28,29], nuclear-gene-based studies consistently position it as sister to subfam. Hippocastanoideae [30,31,32]. This cyto-nuclear discordance is characteristic of patterns often associated with ancient interspecific gene flow [18,20,33,34]. However, this phenomenon has yet to be explored in greater depth.

The primary objective of this study is to leverage comprehensive transcriptome data from diverse Sapindaceae species to identify single-copy orthologs and analyze whole chloroplast genomes, integrating plastid and nuclear datasets to determine whether an ancient interspecific gene flow event involving *Xanthoceras sorbifolium* contributed to the observed cyto-nuclear discordance in its phylogenetic placement. This study aims to clarify the evolutionary relationships within Sapindaceae and explore the role of ancient interspecific gene flow in shaping the family’s diversity.

## 2. Results

### 2.1. Phylogenetic Inference

The integration of 41 RNA-Seq samples alongside 135 nuclear genome samples from the universal Angiosperms353 probe kit resulted in the creation of D2, which encompasses 176 samples representing 123 genera of Sapindaceae. This dataset represents the most comprehensive sampling to date for the phylogenetic analysis of Sapindaceae. Phylogenetic analyses using both concatenated (Appendix A) and coalescent-based methods (Figure 1) yielded largely congruent results, providing strong support for the monophyly of the four subfamilies and their relationships to each other. While the phylogenetic trees are well resolved at the subfamily level, we acknowledge that some nodes within each subfamily exhibit lower support, reflecting uncertainty at finer taxonomic levels. Despite these variations, all Sapindaceae taxa were consistently grouped into four major clades corresponding to the recognized subfamilies. Our analyses support subfam. Dodonaeoideae as the first diverging lineage within Sapindaceae. Additionally, subfam. Xanthoceroideae was resolved as sister to subfam. Hippocastanoideae, and together they formed a clade sister to the largest subfamily, Sapindoideae.

For the smaller dataset D1, we obtained results consistent with D2. Both the concatenation and coalescent-based methods produced a well-resolved phylogeny, with most major clades receiving strong support (Figure 2 and Appendix A). All subsets of D1 recovered the same phylogenetic framework of Sapindaceae, with each of the four subfamilies consistently resolved as monophyletic (Appendix A). Additionally, the relationships among these subfamilies were fully congruent across all datasets: subfam. Dodonaeoideae was the earliest-diverging lineage, while subfam. Xanthoceroideae was recovered as sister to subfam. Hippocastanoideae, and together they formed a clade sister to subfam. Sapindoideae (Figure 2). The only differences among the trees were observed within subfamilies, where some minor branches exhibited variations, such as the precise placement of Matayba within subfam. Sapindoideae. However, these variations did not affect the overall subfamily-level relationships.

For the complete plastid genome data, both ML and Bayesian methods yielded well-resolved phylogenetic trees of Sapindaceae, with only a few clades showing lower support (Figure 3, left). The results obtained under the GTR+G and GTR+G+I models were nearly identical (Figure 3, left; Appendix A). However, the plastid phylogeny exhibited significant incongruence with the species tree derived from nuclear SCOG data (Figure 3, right). Notably, in the plastid phylogeny, subfam. Xanthoceroideae was the first to diverge within Sapindaceae, rather than subfam. Dodonaeoideae. Furthermore, the subfamily Sapindoideae was strongly supported as the sister group to subfam. Dodonaeoideae, with full statistical confidence.

### 2.2. Divergence Time Estimation

The estimation of divergence times indicated that the Sapindaceae family originated approximately 86.9 Mya (95% HPD 80.7–92.6 Mya), while the separation of subfam. Dodonaeoideae from the other three subfamilies took place around 68.6 Mya (95% HPD 63.5–74.0 Mya) during the Late Cretaceous (Figure 2). This divergence was followed by the separation of the ancestor of subfam. Xanthoceroideae and subfam. Hippocastanoideae from that of subfam. Sapindoideae around 62.7 Mya (95% HPD 59.4–66.7 Mya), and the split of subfam. Xanthoceroideae and subfam. Hippocastanoideae occurred at 57.9 Mya (95% HPD 56.5–59.6 Mya) during Paleogene (Figure 2).

### 2.3. Analysis of Gene Tree Discordance

The analysis of conflict in PhyParts v.0.0.1 [35] revealed notable discrepancies throughout the lineages of Sapindaceae, as illustrated in Figure 4. The study validated that the subfam. Sapindoideae is monophyletic, with 95% of the gene trees showing consistent support. Similarly, the monophyly of subfam. Hippocastanoideae was highly supported by 78% of gene trees. Nonetheless, many clades of the Sapindaceae showed considerable levels of gene tree discordance. For instance, the lineages of the subfam. Sapindoideae and subfam. Xanthoceroideae were supported by only 56% and 47% of gene trees, respectively (Figure 4).

To better understand the cyto-nuclear discordance regarding the phylogenetic position of subfam. Xanthoceroideae, we compared the plastid genome tree with simulated nuclear gene trees. Our analysis revealed that only 4% (0.04) of the simulated nuclear gene trees were concordant with the plastid phylogeny (where subfam. Xanthoceroideae was placed at the base of Sapindaceae), indicating that the expected probability of recovering this topology under ILS alone is extremely low (Figure 3). Since ILS alone would theoretically produce a higher proportion of concordant gene trees if it were the primary driver of discordance, these results strongly suggest that ILS alone is insufficient to account for the observed cyto-nuclear conflict.

In contrast, the discordance between nuclear gene trees and the species tree follows a different pattern. By comparing the tree-to-tree distance distributions (Appendix A) obtained from empirical gene trees and simulated datasets, we found that both distributions exhibited similar overall shapes. The most frequently observed tree-to-tree distances were 4 and 6 in both datasets, with nearly identical probabilities, suggesting that ILS is the primary driver of gene tree discordance in Sapindaceae. A Mann–Whitney U test further supported this inference, yielding a *p*-value of 0.30, indicating no significant difference between the empirical and simulated distributions. Additionally, QuIBL analysis reinforced this conclusion, estimating that introgression contributed no more than 5% to gene tree conflicts in Sapindaceae (Appendix A). Collectively, these results demonstrate that ILS is the dominant factor underlying conflicts between nuclear gene trees and the species tree, although other evolutionary processes may have played a minor role.

### 2.4. Assessment of Interspecific Gene Flow

By using HyDe v.1.0 [36] and the visualization script, we observed a notable presence of interspecific gene flow signals within Sapindaceae (Appendix A). Many taxa displayed remarkably complicated gene flow backgrounds, including *Koelreuteria* (Appendix A), *Dodonaea* (Appendix A), and *Xanthoceras* (Appendix A). To focus on interspecific gene flow involving *Xanthoceras*, we excluded other groups that might have undergone gene flow from the HyDe analysis to minimize potential interference with the results. (Appendix A).

The reanalyzed HyDe results (Figure 5A) indicated that *Xanthoceras* may have received an ancient introgression event from subfam. Sapindoideae (Figure 5A), which contributed 16% of its genetic material. The interspecific gene flow involving *Xanthoceras* was further supported by the PhyloNet v.3.8.4 [37] results (Figure 5B and Appendix A). The best-supported number of gene flow events was one (Figure 5B, left).

The STRUCTURE v2.3.4 [38] analysis further corroborated the findings from HyDe and PhyloNet, reinforcing the evidence for genetic admixture in subfam. Xanthoceroideae. The results revealed that the subfam. Xanthoceroideae harbors genetic components from both subfam. Sapindoideae and subfam. Hippocastanoideae. With K = 2, the population structure analysis showed that individuals of *Xanthoceras sorbifolium* exhibit a mixed genetic background rather than a single-source ancestry (Appendix A).

## 3. Discussion

### 3.1. Deciphering Cyto-Nuclear Discordance in Sapindaceae Phylogeny

This study provides a highly resolved phylogeny of Sapindaceae, advancing our understanding of the evolutionary relationships within this diverse family. Consistent with prior research [22,23,24,25,26,27,28,29,30,32], our findings reaffirm the broadly defined Sapindaceae [39]. The four major clades resolved in our analyses align with recent infra-familial classifications [22,30,32], corresponding to the four subfamilies: Dodonaeoideae, Sapindoideae, Xanthoceroideae, and Hippocastanoideae.

A particularly noteworthy finding of this study is the pronounced discrepancy between nuclear and plastid phylogenies, particularly regarding the placement of subfam. Xanthoceroideae. Historically, including in the APG IV system, subfam. Xanthoceroideae has been considered the earliest diverging lineage of Sapindaceae [22,23,24,25,26,27,28,29,39]. However, our results indicate that this topology only holds when inferred from plastid genomes. In contrast, nuclear phylogenies strongly support subfam. Xanthoceroideae as sister to subfam. Hippocastanoideae, a result consistent with recent studies based on nuclear genomic data [30,31,32].

To investigate the cause of cyto-nuclear discordance, we conducted coalescent simulations to assess whether ILS alone could explain the observed conflict. If ILS were the primary driver, a substantial proportion of simulated gene trees would be expected to match the plastid-derived topology [18,19,33]. However, our simulations revealed that only a negligible fraction of nuclear gene trees supported the plastid-based topology for subfam. Xanthoceroideae, strongly suggesting that ILS alone cannot fully account for this discordance.

### 3.2. Detecting Ancient Interspecific Gene Flow Events in the Presence of Strong ILS

Ancient interspecific gene flow can lead not only to cyto-nuclear discordance but also to extensive gene tree-species tree discordance [7,40]. In this study, while we detected widespread conflicts between gene trees and the species tree at several subfamily divergence nodes, coalescent simulations and QuIBL analyses indicated that these conflicts were primarily driven by ILS rather than gene flow. Notably, these subfamilies underwent rapid diversification around the Cretaceous–Paleogene (K–Pg) mass extinction (~65 Mya), a period of accelerated evolutionary radiation that likely led to extensive ILS [20].

To account for ILS while detecting gene flow, we employed HyDe and PhyloNet, both of which integrate ILS into their models [36,37]. Both methods produced consistent results, identifying subfam. Xanthoceroideae as primarily derived from subfam. Hippocastanoideae, with a smaller genetic contribution from subfam. Sapindoideae. These findings do not contradict our coalescent simulations and QuIBL analyses, as gene flow occurring in the presence of strong ILS can lead conservative methods to favor ILS over gene flow as an explanation for discordance.

Additionally, we performed a STRUCTURE analysis despite its known limitations in detecting ancient gene flow [41]. STRUCTURE’s underlying model does not fully account for ILS, as it was not originally designed for introgression detection [41], yet it has been widely applied for this purpose [42]. In our study, STRUCTURE produced results consistent with HyDe and PhyloNet, further supporting the admixture hypothesis for subfam. Xanthoceroideae. However, STRUCTURE also generated false positives, identifying Aesculus and Handeliodendron as admixed despite no supporting evidence from HyDe or PhyloNet. In our case, we speculate that STRUCTURE tends to classify basal lineages of large clades as admixed, potentially leading to misinterpretations. Therefore, we suggest that STRUCTURE and similar programs, such as ADMIXTURE, should be carefully evaluated when applied to interspecific gene flow detection in future studies.

### 3.3. Ancient Introgression of Subfam. Xanthoceroideae

Interspecific gene flow is recognized as a key evolutionary mechanism that has profoundly shaped plant diversity and adaptation [1,2,3,11]. In general, interspecific gene flow can occur through two distinct evolutionary processes: hybridization and introgression. These processes differ in their evolutionary consequences—hybridization typically involves the formation of a new lineage through the merging of two parental genomes, whereas introgression refers to the unidirectional transfer of genetic material from one lineage into another without necessarily leading to speciation [43]. However, distinguishing between these two processes over deep evolutionary timescales remains inherently challenging. In fact, most computational tools designed to detect such events do not explicitly differentiate between hybridization and introgression [36,37], further complicating their resolution in phylogenetic studies.

In the case of subfam. Xanthoceroideae, our HyDe analysis identified two major genetic contributors: subfam. Hippocastanoideae and subfam. Sapindoideae, with highly asymmetric contributions—84% from subfam. Hippocastanoideae and only 16% from subfam. Sapindoideae (Figure 5). Since hybrid speciation typically involves a more balanced genetic contribution from both parental lineages, we infer that such a pronounced disparity suggests that ancient introgression from the ancestral lineage of Sapindoideae into the ancestral lineage of subfam. Xanthoceroideae is a more likely explanation than direct hybridization. However, we acknowledge that repeated backcrossing between a hybrid and one of its parental lineages over time could also lead to an imbalanced genetic contribution, and we cannot entirely rule out this possibility.

Morphological and ecological data further support this evolutionary scenario. subfam. Xanthoceroideae exhibits a mosaic of traits associated with both subfam. Hippocastanoideae and subfam. Sapindoideae, yet the majority of its characteristics closely resembles those of subfam. Hippocastanoideae. For instance, both subfam. Xanthoceroideae and subfam. Hippocastanoideae are predominantly distributed in temperate regions of the Northern Hemisphere, where they have adapted to colder climates, whereas most other Sapindaceae are confined to tropical regions, highlighting a clear ecological distinction [24]. Additionally, the basal lineages of subfam. Hippocastanoideae (e.g., *Billia*, *Handeliodendron*, *Aesculus*) and subfam. Xanthoceroideae share the presence of large, brightly colored flowers, a trait that is rare among other members of Sapindaceae.

Embryological traits further reinforce this pattern. Subfam. Xanthoceroideae and subfam. Hippocastanoideae share key developmental features, including a glandular tapetum, simultaneous microsporogenesis, and two-celled pollen, as well as a floral development pattern in which flowers initially appear bisexual but become functionally unisexual upon maturation [21]. Although subfam. Xanthoceroideae also exhibits certain characteristics found in subfam. Sapindoideae, such as three-colporate pollen grains, axile placentation, and a bitegmic, crassinucellate ovule with a hypostase [21], the number of shared traits with subfam. Hippocastanoideae is substantially greater. This morphological pattern aligns with the genomic evidence supporting introgression from the ancestral lineage of subfam. Sapindoideae into the ancestral lineage of subfam. Xanthoceroideae, further reinforcing the evolutionary scenario inferred from our genetic analyses.

Our study also uncovered other potential instances of introgression within Sapindaceae (Appendix A). For example, while Koelreuteria exhibits no signs of cyto-nuclear discordance, our analyses suggest that it may have acquired genetic material from ancestral lineages of subfam. Hippocastanoideae. Similarly, Dodonaea, which appears as the earliest-diverging lineage in the nuclear phylogeny but is strongly supported as the sister group to subfam. Sapindoideae in the plastid tree, carries genomic signatures indicative of introgression from a lineage outside Sapindaceae, potentially explaining its distinct phylogenetic placement.

Although we cannot precisely determine the timing of these introgression events, divergence estimates suggest that they likely occurred during the Late Cretaceous and Paleogene, coinciding with the Cretaceous–Paleogene (K-Pg) mass extinction—a period of significant ecological upheaval that created numerous new ecological opportunities [44,45]. The occurrence of introgression-driven adaptation during this period may have facilitated the rapid diversification of Sapindaceae, enabling the family to expand into newly available ecological niches.

These findings underscore the critical role of ancient introgression in shaping the early diversification of Sapindaceae, contributing to its complex phylogenetic relationships and morphological diversity. Unlike many angiosperm families where whole-genome duplication (WGD) played a key role in diversification [45,46,47], Sapindaceae lacks evidence of polyploidy [48,49,50,51,52], suggesting that introgression has been a major evolutionary driver. Its remarkable morphological diversity and economically important genera (e.g., Acer, Longan, and Litchi; [53]) further highlight its value as a model for studying gene flow-driven evolution. The unique interplay of introgression, ecological adaptation, and economic significance makes Sapindaceae an ideal system for exploring the broader impacts of introgression on plant diversification.

## 4. Materials and Methods

### 4.1. Data Sources

This study used three different datasets to trace the evolutionary history of the Sapindaceae family, incorporating multiple outgroups from Sapindales and Malvales. The data sources include (D1) single-copy nuclear orthologs (SCOGs) derived from RNA-Seq method, (D2) nuclear genome data from both the Hyb-Seq technique using the universal Angiosperms353 probe kit and the RNA-Seq method, and (D3) complete plastid genome sequences via the genome-skimming approach.

The RNA-Seq dataset (D1) comprises 41 samples representing 12 genera and 25 species within Sapindaceae (Appendix A), encompassing all subfamilies and major lineages of the family, along with eight outgroup taxa from closely related families. Transcriptome reads for each sample were obtained from the Sequence Read Archive (SRA) of GenBank (https://www.ncbi.nlm.nih.gov, accessed on 24 June 2024) and were used for phylogenetic and hybridization analyses. D1 contains a large number of single-copy orthologs (SCOGs), enabling precise ancient hybridization analyses. However, its limited sampling poses potential uncertainties in phylogenetic resolution, such as challenges in confirming the monophyly of the family.

To address these limitations, we incorporated nuclear genome data (D2), primarily sourced from Buerki et al. [30]. D2 includes 135 samples representing 123 of the 144 recognized genera in Sapindaceae s.l. (Appendix A) and was retrieved from the European Nucleotide Archive (https://ebi.ac.uk/ena/browser/home, accessed on 24 June 2024). The RNA-Seq data (D1) were integrated with D2 to provide more comprehensive taxon sampling, encompassing all subfamilies, tribes, and the majority of genera necessary for reconstructing a robust phylogenetic framework of the family (see Appendix A).

The complete plastid genome dataset (D3) included samples from 21 Sapindaceae species and 8 additional outgroup species (Appendix A). This dataset encompassed all subfamilies and major lineages of Sapindaceae and closely corresponded to the sample set from D1. The plastid genome sequences were retrieved from publicly available databases (NCBI) and were used to investigate cyto-nuclear discordance.

### 4.2. Sequence Processing

For the RNA-Seq data (D1), the raw SRA files were extracted using the sratoolkit v. 2.9.2 (https://github.com/ncbi/sra-tools, accessed on 24 June 2024). We applied Trimmomatic v. 0.39 [54] to process the reads and eliminate low-quality bases using the parameters of “LEADING: 3, TRAILING: 3, SLIDINGWINDOW: 4:15, HEADCROP: 8, MINLEN: 36”. Subsequently, the clean RNA-Seq reads were assembled de novo using Trinity v. 2.8.5 [55] with default settings. The longest isoform was selected using a Trinity bundled script “get_longest_isoform_seq_per_trinity_gene.pl”. We applied CD-HIT v4.6.2 [56] in order to reduce redundancy in the raw assemblies. Finally, protein-coding sequences and CDS regions were predicted using TransDecoder v.3.0.1 (https://github.com/TransDecoder, accessed on 24 June 2024).

Protein coding contigs from the 41 samples were subjected to an all-versus-all BLAST analysis using Proteinortho v. 6.0 [57]. We utilized a Python script (https://github.com/Jhe1004/Get_SCOG_from_Proteinortho, accessed on 24 June 2024) to extract single-copy orthologous genes (SCOGs) from the Proteinortho results, ensuring each SCOG was present in at least 28 samples (70% of the total 41 samples). Subsequently, organellar genes were identified and removed via BLAST v. 2.16 (https://blast.ncbi.nlm.nih.gov, accessed on 24 June 2024) analysis against the plastid genomes of an *Acer* species (NC_051542) and a *Dimocarpus* species (OR230532), along with the mitochondrial genome of a *Nelumbo* species (NC030753). The remaining SCOGs were then refined using another Python script (https://github.com/Jhe1004/DelMissingSite, accessed on 24 June 2024), which progressively removed aligned columns containing increasing thresholds of gaps, starting with a 20% “stripping threshold” [58].

Using BLAST to identify orthologous sequences can be challenging due to mis-assembly, deep coalescence, or frame shifts [59]. Additionally, sequences incorrectly clustered as orthologs may exhibit excessively long branches, potentially indicating elevated substitution rates, misalignment, sequencing errors, or model artifacts. To mitigate these issues, we applied TreeShrink v.1.3.9 [60] to detect and remove such outlier sequences. The SCOG sequences were aligned using MAFFT v.7.475 [61] with default parameters, and gene trees were constructed with RAxML v.8.2.12 [62] under the GTR+G model. TreeShrink analysis identified 5925 outlier branches across 3046 genes, which were subsequently removed from the SCOG alignments to reduce potential biases. After filtering, a total of 3326 SCOGs were retained (Appendix A). To further refine the dataset, we applied a 20% “stripping threshold” to remove gaps, resulting in SCOG alignment lengths ranging from 297 to 8571 bp, with an average length of 1315 bp. These steps ensured that the retained sequences were phylogenetically robust and minimized potential errors in downstream analyses.

For D2, the nuclear genome data from the Angiosperms353 probe kit were processed using CD-HIT and TransDecoder to obtain protein-coding genes. These sequences were combined with the 41 samples of the RNA-Seq data using an all-versus-all BLAST analysis implemented in Proteinortho v.6.0. A total of 138 nuclear genes were included in D2, with each gene representing at least 50% of the samples.

The complete plastid genomic sequences were directly downloaded from GenBank (Appendix A). Sequences were then aligned using MAFFT, and a Python script was used to eliminate ambiguous alignments, applying a 20% “stripping threshold”. Finally, we obtained a plastid dataset that has an aligned length of 158,767 bp.

### 4.3. Phylogenetic Analysis

Both the concatenated and coalescent-based methods were applied for phylogenetic reconstruction using the RNA-Seq data (D1). For the concatenation approach, we included 594 SCOGs with less than 10% missing taxa to reduce computation time. These SCOG alignments were concatenated into a super-alignment using the “Concatenate Sequences or Alignments” function in Geneious Prime v. 2024.0.7 [63]. The phylogenetic trees were then reconstructed by RAxML, using the GTR+G model along with 100 bootstrap replicates.

For the coalescent-based methods, we employed both the full-coalescent and a two-step summary coalescent approach. The detailed methodology for the full-coalescent approach is described later in the ’Molecular Dating’ section. The two-step summary coalescent method was implemented using ASTRAL-III v.5.6.3 [64]. All the trees were first generated using RAxML with the GTR+G model. Considering the vast number of gene trees present in this dataset, we took further steps to refine them for better accuracy [65]. We applied two methods for filtering: (1) setting a minimum length for each SCOG alignment [66] and (2) setting a threshold for the average minimum bootstrap values of each gene tree [67]. Applying these two strategies, we obtained five sub-datasets from D1: (1) a total of 3326 SCOGs (SCOG-all), (2) alignments of at least 1000 bp (SCOG-1000bp, 1928 SCOGs), (3) alignments of at least 2000 bp (SCOG-2000bp, 499 SCOGs), (4) trees with at least 70% mean bootstrap values (SCOG-70BS, 2993 SCOGs), and (5) trees with at least 90% mean bootstrap values (SCOG-90BS, 181 SCOGs). All these datasets were used to reconstruct the coalescent species tree with ASTRAL.

The analysis of 176 samples in D2 required substantial computational resources, making both methods highly resource-intensive. To ensure practicality and accuracy in tree-building, the concatenated approach followed the same methodology as in dataset D1, including only 46 out of the 138 nuclear genes, all of which had fewer than 10% missing samples. As the full-coalescent approach was excessively resource-intensive, we adopted the two-step summary method using ASTRAL for species tree reconstruction, following the same workflow as in D1.

For the complete plastid genome data, the aligned sequences were analyzed using both maximum likelihood (ML) and Bayesian methods. The ML analysis was carried out with RAxML under the GTR+G model (with additional calculations using the GTR+G+I model). We assessed the statistical support of each clade with 100 bootstrap replicates. The Bayesian inference (BI) was performed using MrBayes v.3.2.3 [68], implementing the GTR+G model into two independent Markov chain Monte Carlo (MCMC) runs, each consisting of three heated chains and one cold chain, for a total of 2,000,000 generations. Trees were sampled every 100 generations, with the first 25% of sampled trees discarded as burn-in. The remaining trees were used to generate the consensus Bayesian tree.

### 4.4. Molecular Dating

Molecular dating analysis was carried out on D1 using a full-coalescent method implemented in StarBeast3 v.1.1.7 [69]. Because of the high computational demands and the software’s inability to manage missing taxa, we restricted our analysis to the 17 longest SCOG alignments that had no missing taxa to estimate diversification times. The models of sequence evolution for each gene were selected as “GTR+G”, and a “Relaxed Clock Model” was applied for the species tree. The analyses were conducted over 2.0 × 10^8^ generations in the Markov Chain, with the initial 1.0 × 10^8^ generations discarded as burn-in. TreeAnnotator v. 2.7.7 [70] was utilized to visualize the maximum clade credibility phylogeny.

Multiple mega-fossil records were used as the calibrations within Sapindaceae. The stem group age of *Acer*, *Aesculus*, *Dipteronia*, and *Handeliodendron* was constrained to a minimum of 55.8 Mya [24] with a log-normal distribution, based on leaf fossils of *Acer* from the Upper Palaeocene to the Early Miocene [71], fruit fossils of *Dipteronia* from the Palaeocene to the Oligocene [71,72], and leaf and fruit fossils of *Aesculus* from the Palaeocene [72]. The stem group of *Koelreuteria* was constrained to a minimum age of 37.2 Ma [24] with a log-normal distribution, according to leaf and fruit fossils of *Koelreuteria* from the Middle Eocene to Oligocene [71]. We also used a secondary calibration for the stem group age of Sapindales, previously estimated at 97.2 Mya [73] with a normal distribution.

### 4.5. Detecting Gene Tree-Species Tree Discordance

To illustrate conflicts among gene trees, we utilized Toytree v.2.0.5 [74] to generate cloud-tree plots. Since Toytree requires gene trees to be ultrametric and contain a complete set of taxa, we applied the ETE3 Python package v.3.1.2 [75] to filter out gene trees with missing samples from the SCOG-all dataset of D1. As a result, 86 gene trees met these criteria and were retained for visualization. To convert these gene trees into an ultrametric form, we performed time calibration using TreePL v.1.0 [76], applying node calibrations based on age estimates from our dating results via DendroPy v.4.5.2 [77]. This approach provided a qualitative representation of the extent and distribution of topological discordance among gene trees.

We used PhyParts to assess the inconsistencies between gene trees and the species tree. This process involved mapping the nuclear gene trees to the species tree, which was reconstructed using the full coalescent method. We then calculated and summarized the proportion of concordant nodes relative to the combined total of discordant and concordant nodes to provide a precise measure of the level of agreement between the gene trees and the species tree.

### 4.6. Coalescent Simulation for Tree Discordance

Systematic discordances between gene trees and the species tree are commonly attributed to gene flow, ILS, or other sources of error and noise [78]. Among these factors, the relative contributions of introgression and ILS have been widely debated [79]. In this study, we evaluated conflicts between the plastid and species trees, as well as discordances between individual nuclear gene trees and the species tree, aiming to determine whether these conflicts were primarily driven by introgression or ILS.

For nuclear gene-species tree discordance, we conducted coalescent simulations following established methodologies [20,34,80,81]. First, we calculated tree-to-tree distances [82] between each empirical gene tree and the species tree and plotted their frequency distribution. Under the assumption that all observed discordances in the simulated gene trees were exclusively due to ILS, we generated 10,000 gene trees using the multispecies coalescent (MSC) model in the R package Phybase v1.5 [83]. By comparing the frequency distributions of tree-to-tree distances between empirical and simulated gene trees, we assessed the extent to which ILS accounted for the observed discordances. If the two distributions were highly similar, we inferred that ILS was the primary driver of discordance. Conversely, significant differences between the distributions suggested the involvement of additional factors, such as gene flow. Tree-to-tree distances were calculated using DendroPy, and a Mann–Whitney U test was performed to evaluate the null hypothesis that all conflicts resulted solely from ILS. A *p*-value below 0.05 led to the rejection of this hypothesis.

We further employed QuIBL to evaluate the relative contributions of ILS and introgression to nuclear gene-species tree discordance. According to the coalescent model proposed by Edelman et al. [84], internal branches of gene trees for a three-taxon set (triplet) can be modeled as a mixture of two distributions: one representing ILS and the other introgression or speciation events. The estimated mixing proportions (π1 for ILS, π2 for introgression/speciation, where π1 + π2 = 1) reflect the relative influence of each process in shaping gene tree variation. For each triplet, QuIBL calculates the proportions of gene trees supporting the three possible topologies and employs Expectation Maximization (EM) to estimate π1, π2, and other parameters. It also computes Bayesian Information Criterion (BIC) scores to compare models assuming ILS alone versus those incorporating introgression. A high π2 value in cases of species tree discordance suggests a strong signal of introgression. For this analysis, QuIBL was run on each triplet individually with the default parameters (numdistributions: 2; likelihoodthresh: 0.01; numsteps: 50; gradascentscalar: 0.5). Following Edelman et al. [84], we applied a ΔBIC < −30 threshold to identify significant gene flow events. These significant gene flow events were then visualized in a heat map, highlighting the π2 values.

For cyto-nuclear discordance, we employed a similar coalescent simulation to assess whether ILS alone could explain the observed plastid-nuclear conflicts in Sapindaceae [18]. If ILS were the sole contributor, a substantial fraction of simulated gene trees would be expected to match the plastid tree topology. Using Phybase, we generated 10,000 gene trees based on the species tree. We then used Phyparts to identify discrepancies between the simulated trees and the plastid tree, calculating the percentage of simulated gene trees that supported the plastid genome topology.

### 4.7. Introgression Events Detection

We employed HyDe to detect introgression events in Sapindaceae, utilizing a coalescent-based model to infer introgression signals through phylogenetic invariants. Similar to Patterson’s D-statistic [85,86], HyDe operates on a rooted four-taxon tree, structured as “(((P1, H), P2), O),” where “O” represents the outgroup. The model assumes that a proportion γ of the genetic material in “H” has been introgressed from “P2”. Under the null hypothesis of no introgression, γ is expected to be 0. HyDe employs a Z-test to evaluate whether γ significantly deviates from 0, with higher Z-scores indicating stronger statistical support for introgression. In this study, 3326 SCOGs of D1 were concatenated into a super-alignment dataset. Then, HyDe was applied using default parameters, testing all possible combinations of the 40 samples, with *Gossypium hirsutum* as the outgroup. The results underwent significant filtering based on criteria (0 < γ < 1, *p* < 0.05, z > 3). The results of the introgression detection were visualized using a heatmap to clearly display the outcomes. For a detailed explanation of this visualization, please refer to https://github.com/Jhe1004/VisualHyde (accessed on 24 June 2024).

In addition to HyDe, we analyzed phylogenetic networks of Sapindaceae using a pseudolikelihood approach with PhyloNet, a tool designed to detect potential gene flow events under the multispecies coalescent model, allowing direct inference of the gene flow process. To minimize random errors in phylogenetic inference caused by short SCOG alignments [87], we selected the gene trees with an average bootstrap value higher than 90% for this analysis. Due to the high computational demands of PhyloNet, our analysis included only 15 samples, with only 1 sample from each genus kept within Sapindaceae. This analysis allowed zero to ten maximum gene flow events with ten independent runs. The best number of gene flow events was determined by plotting the pseudolikelihood scores. In order to prevent the model from overfitting, a significant rise in the score was anticipated until it achieved the optimal number of gene flow events.

Finally, we employed STRUCTURE to further investigate introgression events between the subfamilies Xanthoceroideae and Sapindoideae. STRUCTURE is a Bayesian clustering method that assigns individuals to genetic clusters based on allele frequencies, allowing for the detection of admixture signals. We analyzed the same dataset of 3326 SCOGs from D1, concatenating the sequences and extracting SNPs for further analysis. To minimize linkage disequilibrium, we selected only the first SNP from every set of ten adjacent SNPs. The aligned sequences were then converted into a STRUCTURE-compatible format using a custom Python script. Following the methodology of Wu et al. [42], we set K = 2 to assess the genetic structure between the two subfamilies. Each run included 100,000 burn-in iterations, followed by 500,000 MCMC iterations to ensure proper convergence. The results were visualized using a Python script, providing a clear graphical representation of population structure and admixture patterns.

## Figures and Tables

**Figure 1 ijms-26-01581-f001:**
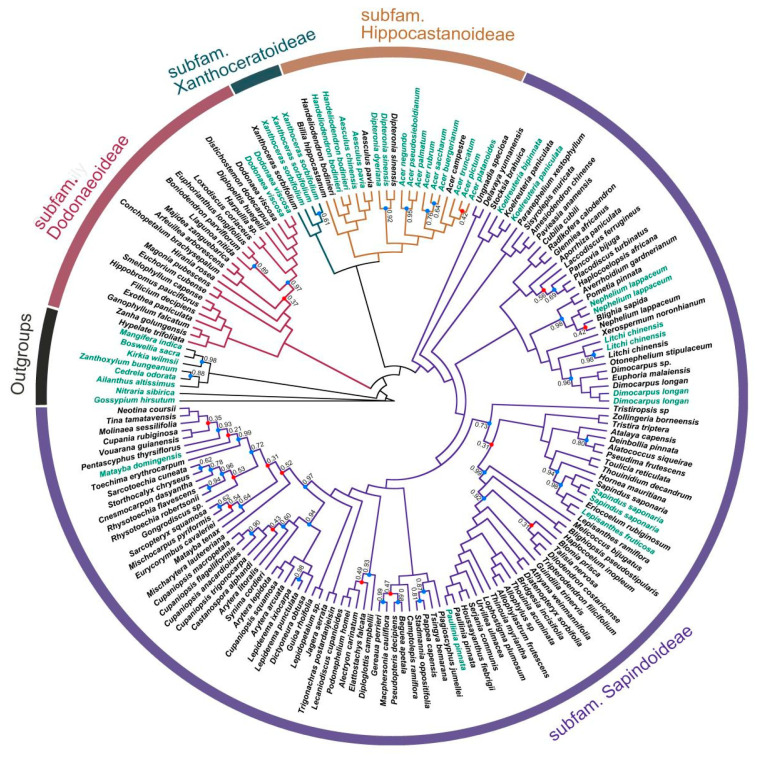
Coalescent-based species tree inference from the Angiosperms353 and transcriptome nuclear data. The topology of the species tree was inferred using ASTRAL based on 138 nuclear genes. The numbers at the nodes denote local posterior probabilities (ASTRAL-lpp), with unmarked nodes indicating a probability of 1.00. The colors of the tree branches correspond to the four subfamilies of Sapindaceae: red for subfam. Dodonaeoideae, green for subfam. Xanthoceroideae, brown for subfam. Hippocastanoideae, and blue for subfam. Sapindoideae. Support values for the branches are indicated by colored circles, where red circles denote nodes with lower support (ASTRAL-lpp < 60%) and blue circles denote nodes with higher support (ASTRAL-lpp ≥ 60%). The sample names are color-coded to indicate data sources, with black labels representing species analyzed using the Angiosperms353 dataset from Buerki et al. [30] and green labels representing species sampled from this study’s transcriptome dataset.

**Figure 2 ijms-26-01581-f002:**
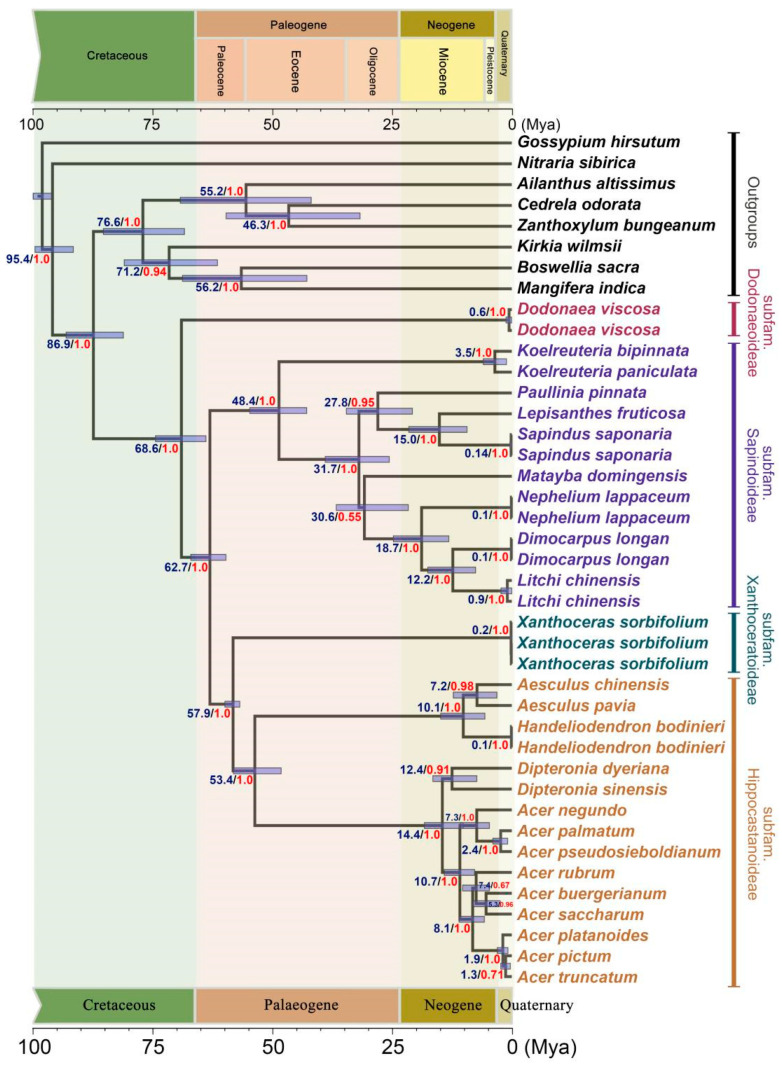
A chronogram of sapindaceae inferred from single-copy orthologous genes using the full-coalescent approach. Blue numbers on the nodes represent the median estimated divergence times in million years ago (Mya), while red numbers indicate the posterior probability support for the topology in the Bayesian inference (Using StarBeast3). Transparent lines over the nodes represent the 95% highest posterior density (HPD) intervals for the divergence times.

**Figure 3 ijms-26-01581-f003:**
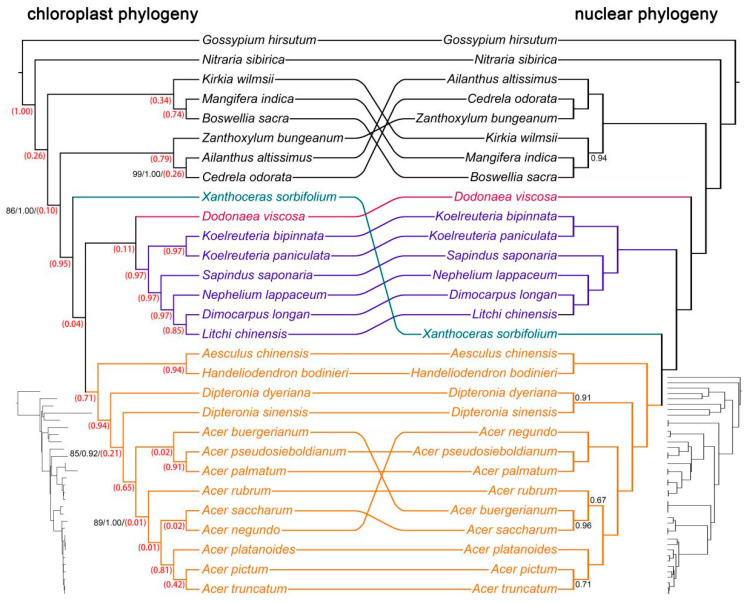
Comparison of the Sapindaceae plastid and nuclear phylogenies. On the left, the Sapindaceae phylogeny, inferred from complete plastid genome sequences using maximum likelihood (ML) with the GTR+G model and Bayesian inference (BI) using MrBayes, is shown. Branches with ML bootstrap percentages less than 100% or BI posterior probability values less than 1 are indicated with the corresponding values. Red numbers in brackets represent the likelihood values indicating the probability of the plastid genome phylogeny topology under scenarios without interspecific gene flow events (considering only incomplete lineage sorting) based on the multispecies coalescent model simulated in Phybase. The ML phylogram is displayed below on the left. On the right, the full coalescence-based species tree topology, inferred using StarBeast3, is presented. Branches with posterior probabilities less than 1.0 are indicated with numbers. The coalescent chronogram is displayed below on the right.

**Figure 4 ijms-26-01581-f004:**
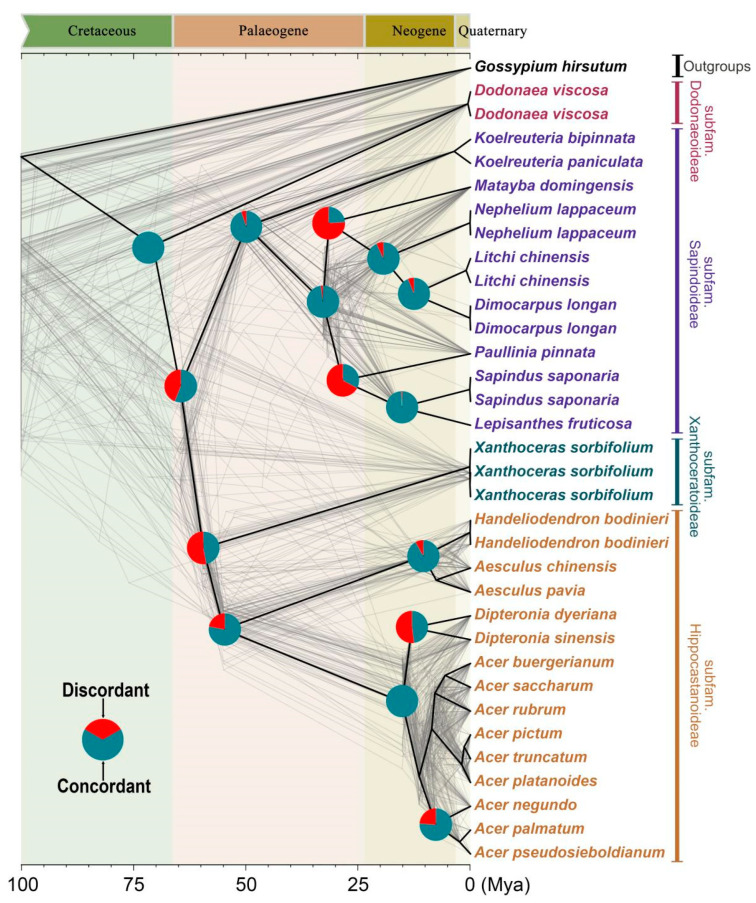
Comprehensive full coalescence-based species tree topology illustrating gene discordance. The species tree is depicted with heavy black lines. The gray-colored trees (cloud-tree) were sampled from 86 single-copy orthologous nuclear genes (excluding those with missing taxa) and constructed using RAxML. Pie charts at each node display the proportions of tree topologies that are concordant and discordant with the overarching species tree, providing a visual quantification of phylogenetic agreement and conflict.

**Figure 5 ijms-26-01581-f005:**
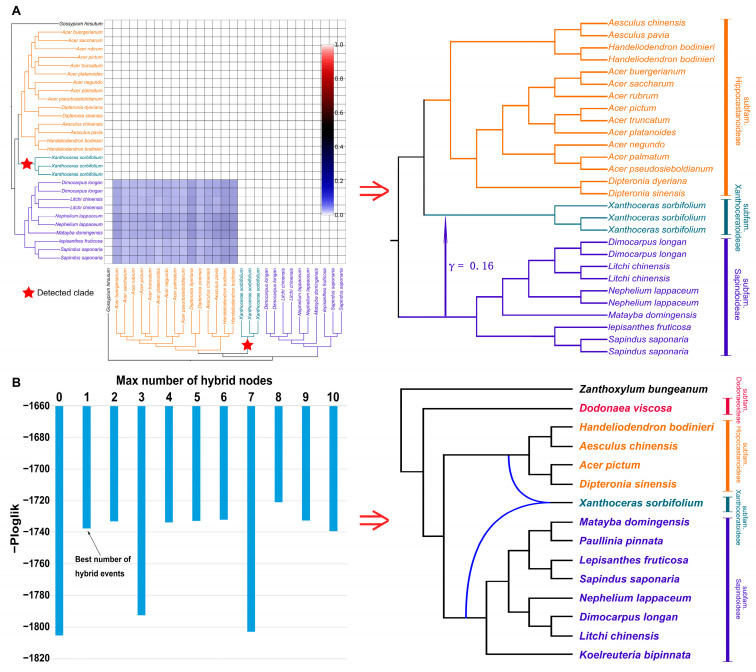
Interspecific gene flow detection based on HyDe (**A**) and PhyloNet (**B**) results. (**A**) On the left, the detected clade (*Xanthoceras*) is marked with a red star on the cladogram. The small blue squares in the heatmap indicate the corresponding interspecific gene flow signals detected by HyDe. The intensity of the blue color represents the inheritance probabilities of the corresponding taxa on the left axis. When small squares of the same color intensity form a larger square in the heatmap, and their corresponding taxa on the coordinate axis form a monophyletic clade, it suggests that the most recent common ancestor of the clade may be one of the parents. On the right, a schematic diagram of a phylogenetic network is shown based on the left cladogram and heatmap. Numerical values indicate inheritance probabilities. (**B**) Interspecific gene flow within Sapindaceae clades as estimated by PhyloNet. The results show maximum pseudolikelihood trees (right), with one best-allowed reticulation based on the pseudo-log-likelihood scores (left).

## Data Availability

The data that support the findings of this study are openly available in the Science Data Bank at 10.5281/zenodo.13150091.

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
