# Peer review of "Unraveling the Ancient Introgression History of Xanthoceras (Sapindaceae): Insights from Phylogenomic Analysis"

_ijms, 2025, doi:10.3390/ijms26041581_

Round 1
Reviewer 1 Report
Comments and Suggestions for Authors
1.In the methods section, many software references are repeated multiple times, such as “RAxML v.8.2.12 (Stamatakis, 2014)”, “MAFFT v.7.221 (Katoh and Standley, 2013)”, and “ASTRAL-III v.5.6.3 (Zhang et al., 2018)”. Please check the entire text and retain the citations only at their first occurrence. Additionally, there is inconsistency in the citation format within the manuscript, with instances of both “and” and “&”. Please make uniform modifications according to the journal's requirements.
2.The version information for Geneious Prime software is missing.
3.The term “incomplete lineage sorting (ILS)” appears multiple times in the manuscript; please retain only its first occurrence.
4.In the methods section regarding the plastome phylogenetic tree, the RAxML software uses the GTR+G model. Why was the GTR+G model chosen? Was this based on previous literature, or is it necessary for this study? In past research, the GTR+G+I model (denoted as GTRGAMMAI in the software) has been more commonly used for plastome data.
5.The manuscript primarily studies the nuclear genome and plastome. I would like to know if the cyto-nuclear discordance observed in Xanthoceras occurs only between the plastome and the nuclear genome? Is there also a conflict between the mitochondrial data and the nuclear genome?
Author Response
We sincerely appreciate the reviewer’s insightful comments and constructive feedback, which have greatly helped us improve the clarity, consistency, and methodological rigor of our manuscript. In response, we have carefully revised the manuscript to address all concerns, including:
- Clarifying methodological choices and ensuring consistency in software citations and formatting.
- Refining phylogenetic analyses, including an additional GTR+G+I model test for plastome phylogeny.
- Improving clarity and readability, including revisions to terminology and figure legends.
We are grateful for the reviewer’s valuable input, which has significantly strengthened the manuscript. Below, we provide detailed responses to each comment.
Comment 1:
In the methods section, many software references are repeated multiple times, such as “RAxML v.8.2.12 (Stamatakis, 2014)”, “MAFFT v.7.221 (Katoh and Standley, 2013)”, and “ASTRAL-III v.5.6.3 (Zhang et al., 2018)”. Please check the entire text and retain the citations only at their first occurrence. Additionally, there is inconsistency in the citation format within the manuscript, with instances of both “and” and “&”. Please make uniform modifications according to the journal's requirements.
Response 1:
Thank you for your careful review and for pointing out these inconsistencies. We have revised the Methods section to ensure that software references are cited only at their first occurrence, eliminating unnecessary repetitions. Additionally, we have standardized the citation format throughout the manuscript to align with the journal’s requirements, ensuring consistency in the use of “and” vs. “&” where applicable. We appreciate your attention to these details, which has helped improve the clarity and professionalism of our manuscript.
Comment 2:
The version information for Geneious Prime software is missing.
Response 2:
Thank you for your comment. We have added the missing version information for Geneious Prime (v.2024.0.7) in the Methods section to ensure completeness and accuracy. We appreciate your attention to this detail.
Comment 3:
The term “incomplete lineage sorting (ILS)” appears multiple times in the manuscript; please retain only its first occurrence.
Response 3:
Thank you for your suggestion. We have revised the manuscript to ensure that “incomplete lineage sorting (ILS)” is defined only at its first occurrence, with subsequent mentions using only the abbreviation “ILS” for consistency. We appreciate your attention to detail, which has helped improve the clarity and readability of the manuscript.
Comment 4:
In the methods section regarding the plastome phylogenetic tree, the RAxML software uses the GTR+G model. Why was the GTR+G model chosen? Was this based on previous literature, or is it necessary for this study? In past research, the GTR+G+I model (denoted as GTRGAMMAI in the software) has been more commonly used for plastome data.
Comment 4:
Thank you for your insightful suggestion. In response, we have conducted an additional plastome phylogenetic analysis using the GTR+G+I model to evaluate its impact on tree topology and support values. The results indicate that the phylogenetic tree constructed with GTR+G+I is topologically identical to the tree inferred using GTR+G, with similar branch support values. The tree generated under GTR+G+I has been included in Figure S8 for reference. We appreciate your suggestion, which has helped strengthen the methodological rigor of our study.
Comment 5:
The manuscript primarily studies the nuclear genome and plastome. I would like to know if the cyto-nuclear discordance observed in Xanthoceras occurs only between the plastome and the nuclear genome? Is there also a conflict between the mitochondrial data and the nuclear genome?
Comment 5:
Thank you for your insightful comment. Investigating mitochondrial-nuclear discordance could indeed be a valuable research direction. In this study, we attempted to extract mitochondrial gene data from our transcriptome dataset and constructed a mitochondrial phylogenetic tree (included in the supplementary material of this response). However, we found that the resulting tree had very low branch support, leading us to exclude it from the main manuscript after careful consideration. Despite the low resolution, we observed that the overall topology of the mitochondrial tree closely resembles that of the plastome tree, with Xanthoceras positioned at the base of Sapindaceae, which contrasts significantly with the nuclear phylogeny. The low resolution of the mitochondrial tree is likely due to the slow evolutionary rate of mitochondrial genes, resulting in an insufficient number of informative sites for reconstructing deeper phylogenetic relationships. While the inclusion of non-coding mitochondrial regions might provide additional phylogenetic signals, this approach is complicated by the high frequency of gene order rearrangements in plant mitochondrial genomes. Such extensive structural variation makes it challenging to reliably align non-coding regions for phylogenetic inference at the intergeneric or higher taxonomic levels, such as those explored in our study.
We appreciate the reviewer’s suggestion and acknowledge that further research incorporating mitochondrial genome-wide data could provide additional insights into cyto-nuclear discordance in Xanthoceras and Sapindaceae as a whole.

Reviewer 2 Report
Comments and Suggestions for Authors
The authors of the current study utilized concatenated and coalescence-based phylogenies with transcriptomic data from selected species of Sapindaceae to investigate the evolutionary history of Xanthoceras in relation to its sister groups. By examining discordance between nuclear gene trees and the plastid genome, the authors conclude that Xanthoceras originated through ancient homoploid hybridization between Hippocastanoideae and Sapindoideae, two subfamilies within Sapindaceae. While the authors provide compelling evidence supporting the hybridization hypothesis, their argument against alternative explanations such as incomplete lineage sorting (ILS) and repeated introgression requires further attention. In particular, additional analyses are needed to rigorously rule out introgression as a driving factor.
Major Concerns
1. Need for Stronger Evidence Against Introgression
• The study does not include sufficient tests to rule out introgression. Given the availability of genome-wide data, why not incorporate analyses such as ADMIXTURE or STRUCTURE to quantify genetic ancestry more explicitly?
• A discussion on the likelihood of introgression should be expanded. How can introgression be definitively excluded from the observed patterns of discordance?
• The conclusions regarding hybrid origin are too strong. A more cautious interpretation is warranted, as genome-scale analyses would be necessary to definitively confirm hybridization.
• Results from HyDe show skewed proportions of genetic contribution, which appear to support introgression rather than hybridization. This should be addressed more explicitly.
Results Section
• The authors state that phylogenetic trees are “highly resolved,” but this seems like an overstatement. Several major clades exhibit low node support across different datasets, which weakens the claim of resolution.
• Figure 1: The meaning of different color labels and node colors (e.g., blue vs. red) is unclear. These should be clarified in the figure legend.
• Paragraph 2: The authors suggest that differing sequence lengths and bootstrap support values influence tree topology. However, why wouldn’t these factors generate the same topology under similar conditions?
• Page 7, last paragraph: The probability of the presented tree topology under ILS alone is reported as 0.95, which is quite high. This suggests that the hybridization hypothesis is not the only plausible explanation. Could this be an indication that ILS alone can account for the observed discordance?
• Page 8, Section 2.4: The word “remarkably” is misspelled.
Discussion Section
• Page 9, third paragraph: The conclusion regarding hybridization is too strong, considering that some analyses were limited by sample size and certain phylogenetic trees support ILS. The discussion should acknowledge these limitations.
• Page 10, first paragraph: Are there any quantitative morphological traits that correlate with the genetic evidence for hybridization? Including such data would strengthen the conclusions.
Materials and Methods
• Page 11, third paragraph: The phrase “dataset (D3) comprised” needs grammatical correction. Additionally, the last sentence in this section should be rephrased for clarity.
• Page 11, last paragraph: What is meant by “unusually long branches”? Does this refer to elevated substitution rates, model artifacts, or something else?
• Page 12, paragraph 5: The last two sentences are repeated from the results section. These should either be removed from this section or omitted in the results to avoid redundancy.
• Page 13, paragraph 4: The dataset appears to shrink progressively during analysis. The final 86 gene trees used for discordance analysis might be insufficient to detect localized introgression.
• A more explicit justification for the adequacy of the dataset is needed.
• Providing an approximate proportion of genes analyzed relative to the whole genome would help readers assess the robustness of the study.
Author Response
We sincerely appreciate the reviewer’s constructive comments and insightful suggestions, which have greatly improved the clarity and rigor of our manuscript. In response, we have made several key revisions:
- Substantial revisions to the Discussion section – We have softened the hybridization claim, reframing the event as introgression and providing a more detailed discussion on the distinction between the two processes. Additionally, we conducted more extensive coalescent simulations and QuIBL analyses to further evaluate these factors and offered a nuanced discussion of ILS and alternative explanations.
- Further validation using STRUCTURE – We performed a STRUCTURE analysis to further assess gene flow patterns, which corroborated the findings from HyDe and PhyloNet. However, we also discuss the limitations and potential false positives associated with STRUCTURE in Section 3.2.
- Clarification of key methodological aspects – We have revised the Results and Methods sections to clarify dataset usage, particularly distinguishing the 86 gene trees used in Toytree analysis from the 3,326 SCOGs used for introgression detection.
- Enhanced figure and text clarity – We have refined Figure 1 and its legend, corrected minor errors, and reworded ambiguous statements throughout the manuscript to improve clarity and readability.
Below, we provide detailed responses to each of the reviewer’s comments.
Major Concerns
Comment 1:
- Need for Stronger Evidence Against Introgression
- The study does not include sufficient tests to rule out introgression. Given the availability of genome-wide data, why not incorporate analyses such as ADMIXTURE or STRUCTURE to quantify genetic ancestry more explicitly?
- A discussion on the likelihood of introgression should be expanded. How can introgression be definitively excluded from the observed patterns of discordance?
- The conclusions regarding hybrid origin are too strong. A more cautious interpretation is warranted, as genome-scale analyses would be necessary to definitively confirm hybridization.
- Results from HyDe show skewed proportions of genetic contribution, which appear to support introgression rather than hybridization. This should be addressed more explicitly.
Response 1:
- Thank you for your thoughtful comments. We have carefully reassessed the classification of the gene flow event involving subfam. Xanthoceroideae and agree that introgression is a more precise term than hybridization in this context. This conclusion is based on the highly asymmetric genetic contributions detected in our HyDe analysis (84% from Hippocastanoideae and only 16% from Sapindoideae) and the limited taxon sampling in our study. However, we acknowledge that, given the deep evolutionary timescale of this event, it remains challenging to definitively distinguish between hybridization and introgression, even with whole-genome data. Highly asymmetric genetic contributions can also arise from repeated backcrossing between a hybrid and one of its parental lineages, further complicating the distinction. Additionally, genomic signatures commonly used to infer introgression, such as synteny patterns and chromosomal structures, may no longer be reliable over such long evolutionary timescales due to extensive genomic rearrangement and lineage-specific evolution. Thus, providing conclusive evidence to distinguish between these two processes is inherently difficult, a limitation we explicitly acknowledge in our manuscript.
- Notably, the HyDe (DOI: 10.1093/sysbio/syy023) and PhyloNet (DOI: 10.1186/1471-2105-9-322) software used in our study does not explicitly differentiate hybridization from introgression—their authors classify all gene flow events under the broader term “hybridization.” In our initial manuscript, we followed this convention; however, upon further consideration, we now adopt introgression as the more precise terminology. To clarify this distinction, we have expanded Section 3.3 to discuss this issue in greater detail.
- We also appreciate the reviewer’s suggestion to use STRUCTURE to further validate our gene flow results. In response, we conducted a STRUCTURE analysis, which confirmed the conclusions derived from HyDe and PhyloNet. The relevant results have been incorporated into the manuscript. However, we also recognize the limitations and criticisms associated with STRUCTURE and similar programs, particularly their inability to account for ILS and other evolutionary processes in their model assumptions, as extensively discussed in 10.1093/sysbio/syaa092. While STRUCTURE successfully identified hybrid taxa, we also observed false positives, particularly the misclassification of basal lineages of large clades as admixed. This suggests that, while STRUCTURE can serve as a useful validation tool, caution is required when interpreting its results as definitive evidence of hybridization. These issues have been carefully discussed in Section 3.2, where we provide a detailed evaluation of the strengths and limitations of this approach.
- To enhance clarity and accuracy, we have substantially rewritten the Discussion section to better reflect these considerations. We sincerely appreciate the reviewer’s insightful suggestions, which have significantly improved the clarity, depth, and rigor of our discussion.
Results Section
Comment 2:
- The authors state that phylogenetic trees are “highly resolved,” but this seems like an overstatement. Several major clades exhibit low node support across different datasets, which weakens the claim of resolution.
Response 2:
- Thank you for pointing out this important issue. Our statement regarding the phylogenetic trees being “highly resolved” specifically referred to the monophyly of the four subfamilies within Sapindaceae and their relationships to each other, which were consistently well-supported. However, we acknowledge that within each subfamily, several nodes exhibit lower support, indicating limited resolution at finer taxonomic levels. While the primary focus of this study is on the relationships among the four subfamilies, we agree that our wording should more accurately reflect the varying levels of resolution throughout the tree. We have revised the manuscript accordingly to clarify this distinction. Thank you again for your valuable feedback.
Comment 3:
- Figure 1: The meaning of different color labels and node colors (e.g., blue vs. red) is unclear. These should be clarified in the figure legend.
Response 3:
- Thank you for your valuable feedback regarding Figure 1. We appreciate your suggestion to clarify the meaning of different color labels and node colors in the figure legend. In this figure, the different branch colors represent the four subfamilies of Sapindaceae: red for subfam. Dodonaeoideae, green for subfam. Xanthoceratoideae, brown for subfam. Hippocastanoideae, and blue for subfam. Sapindoideae. Additionally, the red and blue circles on the tree indicate node support values, where red circles represent nodes with lower support (local posterior probability < 60%) and blue circles represent nodes with higher support (local posterior probability > 60%). The sample names are also color-coded to indicate data sources: black labels correspond to species analyzed using the Angiosperms353 dataset from Buerki et al. (2021), while green labels indicate species sampled from this study’s transcriptome dataset. We have revised the figure legend accordingly to include these explanations. Thank you again for your helpful suggestion, which has improved the clarity of our figure.
Comment 4:
- Paragraph 2: The authors suggest that differing sequence lengths and bootstrap support values influence tree topology. However, why wouldn’t these factors generate the same topology under similar conditions?
Response 4:
- Thank you for this insightful comment. We acknowledge that our wording may have caused some misunderstanding. In fact, across all four datasets generated through gene tree filtering, as well as the dataset that includes all gene trees, the resulting species trees exhibit highly consistent topologies. The only differences among these trees occur in a few minor branches, whereas the monophyly of the four subfamilies and their relationships to each other remain entirely unchanged.As for why these minor topological differences exist, we believe they are primarily influenced by introgression events. The species tree inference in our study was conducted using a multi-species coalescent (MSC) model, which assumes no introgression between lineages, including both sampled taxa and their ancestors. However, given that introgression likely occurred in the evolutionary history of Sapindaceae, this assumption is violated to some extent. As a result, different input datasets may lead to minor discrepancies in inferred topologies due to model limitations.Nevertheless, these small-scale topological conflicts do not affect the core conclusions of our study. The relationships among the four subfamilies remain fully consistent, ensuring the robustness of our findings. We have clarified this point in the revised manuscript. Thank you again for your valuable feedback.
Comment 5:
- Page 7, last paragraph: The probability of the presented tree topology under ILS alone is reported as 0.95, which is quite high. This suggests that the hybridization hypothesis is not the only plausible explanation. Could this be an indication that ILS alone can account for the observed discordance?
Response 5:
- Thank you for your insightful comment. The misunderstanding likely arose due to a lack of clarity in our original description. We would like to clarify that the probability of the presented tree topology under ILS alone is 0.04, not 0.95, indicating that only a very small proportion of simulated nuclear gene trees were concordant with the plastid phylogeny. This suggests that ILS alone is unlikely to account for the observed cyto-nuclear discordance. While ILS may still play a role, the extremely low probability derived from our simulations strongly supports historical introgression events as a more plausible explanation for the observed conflict. To address this, we have revised the manuscript to provide a clearer explanation and avoid any potential confusion.
Comment 6:
- Page 8, Section 2.4: The word “remarkably” is misspelled.
Response 6:
• Thank you for catching this typo. We have corrected “remarkabley” to “remarkably” in the revised manuscript.
Discussion Section
Comment 7:
- Page 9, third paragraph: The conclusion regarding hybridization is too strong, considering that some analyses were limited by sample size and certain phylogenetic trees support ILS. The discussion should acknowledge these limitations.
Response 7:
- Thank you for your valuable comment. Based on your feedback, we have substantially revised this section of the discussion (now reflected in Sections 3.1 and 3.2) to soften the conclusion regarding hybridization and instead frame the event as introgression. In the revised text, we provide a more detailed rationale for why introgression is the more likely explanation, while also acknowledging the uncertainties and limitations of our conclusions. We recognize that our previous wording was too strong, and we now explicitly discuss the role of ILS as a possible alternative explanation, as well as explore why such extensive ILS may have occurred. We appreciate your constructive feedback, which has significantly improved the clarity and balance of our discussion.
Comment 8:
- Page 10, first paragraph: Are there any quantitative morphological traits that correlate with the genetic evidence for hybridization? Including such data would strengthen the conclusions.
Response 8:
• Thank you for your insightful comment. We acknowledge that the morphological traits we discuss are limited and that obtaining meaningful quantitative morphological data is particularly challenging in this case. This difficulty arises because the introgression event in question (from subfam. Sapindoideae into subfam. Xanthoceroideae) occurred at least 48 million years ago, based on our estimates. Since then, these lineages have undergone long periods of independent evolution, which may have obscured direct morphological correlations with genetic introgression signals. However, in response to this suggestion, we have expanded Section 3.3 to incorporate additional macromorphological and ecological details that further support the inferred introgression event. We appreciate this constructive feedback, which has helped improve the depth of our discussion.
Materials and Methods
Comment 9:
- Page 11, third paragraph: The phrase “dataset (D3) comprised” needs grammatical correction. Additionally, the last sentence in this section should be rephrased for clarity.
Response 9:
- The grammatical issue with “dataset (D3) comprised” has been corrected in the revised manuscript by replacing it with “included.” Additionally, the last sentence has been rephrased for clarity, specifying the data source (NCBI) and improving readability. These changes have been incorporated into the updated version.
Comment 10:
- Page 11, last paragraph: What is meant by “unusually long branches”? Does this refer to elevated substitution rates, model artifacts, or something else?
Response 10:
- The term “unusually long branches” in this context refers to sequences that exhibit excessive divergence compared to the overall gene tree topology. These long branches may result from elevated substitution rates, alignment errors, sequencing artifacts, mis-assembly, or incorrect orthology inference. TreeShrink detects such branches by identifying sequences that are outliers in terms of branch length relative to the overall distribution in the tree. These outliers often indicate potential errors rather than true biological variation, which is why they are removed to improve the accuracy of downstream phylogenetic analyses. We have now provided a more detailed explanation in the revised manuscript to clarify this point.
Comment 11:
- Page 12, paragraph 5: The last two sentences are repeated from the results section. These should either be removed from this section or omitted in the results to avoid redundancy.
Response 11:
• Thank you for your feedback. To avoid redundancy, we have revised this section to clarify that the concatenated approach follows the same methodology as in dataset D1. The repeated details about sequence concatenation and tree reconstruction have been removed accordingly.
Comment 12:
- Page 13, paragraph 4: The dataset appears to shrink progressively during analysis. The final 86 gene trees used for discordance analysis might be insufficient to detect localized introgression.
- A more explicit justification for the adequacy of the dataset is needed.
- Providing an approximate proportion of genes analyzed relative to the whole genome would help readers assess the robustness of the study.
Response 12:
- Thank you for your thoughtful comments and for highlighting the need for further clarification. The 86 gene trees used in this section were not intended for introgression detection or hybridization analysis. Instead, they were specifically selected to assess the degree of conflict between gene trees and the species tree using cloud-tree plots. The substantial reduction in the number of gene trees was due to the requirements of Toytree, which necessitates that each gene tree be ultrametric and complete, meaning all taxa must be present. As a result, only 86 gene trees from the SCOG-all dataset met this criterion. This analysis was meant to provide a qualitative visualization of topological discordance rather than a quantitative assessment of introgression.To improve clarity, we have revised the corresponding section of the manuscript to ensure that our intent and methodology are more explicitly conveyed.
- For introgression detection, we used 3,326 gene trees in the analysis presented inSection 4.7, ensuring a comprehensive evaluation. We apologize for any confusion and have now revised the manuscript to clearly distinguish the purpose of this analysis and justify the adequacy of the dataset. Thank you again for your valuable feedback.